# Numerical Analysis on Natural Convection Heat Transfer in a Single Circular Fin-Tube Heat Exchanger (Part 2): Correlations for Limiting Cases

**DOI:** 10.3390/e22030358

**Published:** 2020-03-20

**Authors:** Jong Hwi Lee, Young Woo Son, Se-Myong Chang

**Affiliations:** Department of Mechanical Engineering, Kunsan National University, Gunsan 54150, Korea; jhl@kunsan.ac.kr (J.H.L.); syw@kunsan.ac.kr (Y.W.S.)

**Keywords:** natural convection, circular fin-tube, heat exchanger, correlations

## Abstract

This research focused on the correlations associated with the physics of natural convection in circular fin-tube models. The limiting conditions are defined by two conditions. The lower limit ($D_{o}/D$ → 1, *s*/*D* = finite value) corresponds to a horizontal circular tube, while the upper limit ($D_{o}/D$ → ∞, *s*/*D* << 1) corresponds to vertical flat plates. In this paper, we proposed a corrected correlation based on empirical result. The circular fin-tube heat exchanger was divided into the A and B types, the categorizing criteria being $D_{o}/D=1.2$, where D and $D_{o}$ refer to the diameter of the circular tube and the diameter of the circular fin, respectively. Moreover, with the computational fluid dynamics technique used to investigate the limiting conditions, the parametric range was extended substantially in this research for type B, namely 1.2 < $D_{o}/D$ ≤ 10. The complex correlation was also simplified to the form ${\mathrm{Nu}}_{L}$= $C{\mathrm{Ra}}_{s}{}^{n}$, where C and n are the functions of the diameter ratio $D_{o}/D$.

## 1. Introduction

In Part 1 [1] of this paper, a numerical method was proposed for the natural convection heat transfer in circular fin-tube heat exchangers. The results validate the accuracy of the proposed computation method and its ability to serve as a suitable design reference. In this paper (Part 2), our objective was to study the relevant correlations to better understand the physics of natural convection in circular fin-tube models. Thus far, many studies have been published on this topic, with some representative ones focusing on the following correlations.

The simple classical corre1ation for natural convection in a circular tube or sphere was proposed in 1954 by Merk and Prins [2].
(1)$${\mathrm{Nu}}_{D}=C{\mathrm{Ra}}_{D}{}^{\frac{1}{4}},C=0.436,\;\mathrm{Pr}=c_{p}\mu /k=0.7.$$

In 1975, Morgan [3] revised the correlation in Equation (1) by tabulating C and n, which are functions of the Rayleigh number (see Table 1).
(2)$${\mathrm{Nu}}_{D}=C{\mathrm{Ra}}_{D}{}^{n}.$$

In 1975, Churchill and Chu [4,5] extended the validity of these correlations to not only a horizontal circular cylinder, but also a flat plate for both laminar and turbulent flow regimes. The following equation shows the correlation for a circular cylinder, while the characteristic length is corrected for a vertical flat plate.
(3)$${\mathrm{Nu}}_{D}={\left\{0.6+\frac{0.387{\mathrm{Ra}}_{D}^{\frac{1}{6}}}{{\left[1+{\left(\frac{0.559}{\mathrm{Pr}}\right)}^{\frac{9}{16}}\right]}^{\frac{8}{27}}}\right\}}^{2}\;\mathrm{and}$$
(4)$$\mathrm{Pr}=0.7,{\;\mathrm{Nu}}_{D}={\left(0.6+0.321{\mathrm{Ra}}_{D}^{\frac{1}{6}}\right)}^{2}.$$

In 1979, Fujii et al. [6] performed numerical analyses on an isothermal horizontal cylinder. The relevant range was found to be $10^{-4}\leq {\mathrm{Gr}}_{D}\leq 10^{4}$ (${\mathrm{Gr}}_{D}={\mathrm{Ra}}_{D}/\mathrm{Pr}$), for Pr values of 0.7, 10, and 100.
(5)$$\frac{2}{{\;\mathrm{Nu}}_{D}}=\ln\left[1+\frac{4.065}{C\left(\mathrm{Pr}\right){\mathrm{Ra}}_{D}^{m}}\right],$$
(6)$$m=\frac{1}{4}+\frac{1}{10+4{\mathrm{Ra}}_{D}^{\frac{1}{8}}},\;\mathrm{and}$$
(7)$$C\left(\mathrm{Pr}\right)=\frac{0.671}{{\left[1+{\left(\frac{0.492}{\mathrm{Pr}}\right)}^{\frac{9}{16}}\right]}^{\frac{4}{9}}}.$$

All the correlations in Equations (1)–(5) use the diameter (*D*) of a circular tube as a reference characteristic length. However, to apply these types of correlations to circular fin-tube heat exchangers, serious consideration must be given to the selection of the characteristic length with regard to Nu and Ra.

In 2011, Kang and Jang [7] proposed a correlation for circular fin-tube heat exchangers using parametric ranges of $3500\leq {\mathrm{Ra}}_{D}\leq 8\times 10^{5}$, 1.6$<D_{o}/D<3.0$, and 0.19$<P_{f}/D<0.34$, where $D_{o}$ is the outer diameter of the fin, and $P_{f}$ refers to the fin pitch.
(8)$${\mathrm{Nu}}_{D}=0.3+2.75{\mathrm{Ra}}_{D}{}^{0.25}{\left(\frac{D_{o}}{D}\right)}^{-1.09}{\left(\frac{P_{f}}{D}\right)}^{0.95}.$$

In 2016, Chen et al. [8] performed a series of experiments on various tube diameters and fin pitches for vertical fin-tube heat exchangers, and presented the following correlations using the least-squares fitting method:(9)$${\mathrm{Nu}}_{s}=-1.432+1.412{\mathrm{Ra}}_{s}{}^{0.25}\;(\mathrm{for}\;\mathrm{the}\;\mathrm{non}\text{-}\mathrm{isothermal}\;\mathrm{situation})$$
(10)$${\mathrm{Nu}}_{s}=-0.516+0.667{\mathrm{Ra}}_{s}{}^{0.25}\;(\mathrm{for}\;\mathrm{the}\;\mathrm{isothermal}\;\mathrm{situation})$$

Recently, Kang and Chang [9] decoupled the physics of natural convection in a circular fin-tube heat exchanger from natural convection in a horizontal cylinder and vertical parallel plates, and defined the limiting conditions (see Figure 1) from their experimental research. Therefore, the natural convection of a fin-tube exchanger can be regarded as a blend of the following two limiting cases. *D_o_*/*D* → 1 and *s*/*D* are *finite* values for a single horizontal cylinder, where Nu is proportional to the power 0.25 of Ra, whereas when *D_o_*/*D* → ∞ and *s*/*D* << 1 for vertical parallel plates, Nu is proportional to the power unity of Ra. They studied 16 types of heat exchangers, with the diameter ratios (*D_o_/D*) ranging from 1.2 to 2.8, and the normalized gap of the fins (*s/D*) ranging from 1.2 to 2.6. The following correlation was finalized:
(11)$$\begin{array}{c} {\mathrm{Nu}}_{L}={\left\{{\left[{\left(a\frac{D_{o}}{D}\right)}^{-b}\right]}^{n}+{\left[\left\{\left[\frac{\pi }{2}C{\left(\frac{s}{D}\right)}^{-\frac{3}{4}}-\frac{1}{6\pi }\right]{\left(\frac{D_{o}}{D}\right)}^{-e}+\frac{1}{6\pi }\right\}{\left(\frac{s}{D_{o}}\right)}^{\tilde{f}}{\mathrm{Ra}}_{s}{}^{{\tilde{g}}_{mod}}\right]}^{n}\right\}}^{\frac{1}{n}}\\ =\{{\left[{\left(a\frac{D_{o}}{D}\right)}^{-0.58}\right]}^{3}+[\{\left[0.685{\left(\frac{s}{D}\right)}^{-0.75}-0.053\right]{\left(\frac{D_{o}}{D}\right)}^{-0.27}+\\ 0.053\}{\left(\frac{s}{D_{o}}\right)}^{1-{\left(\frac{D_{o}}{D}\right)}^{-0.57}}{\mathrm{Ra}}_{s}{}^{\left[1-0.75{\left(\frac{D_{o}}{D}\right)}^{-0.3\eta }\right]}{]}^{3}{\}}^{\frac{1}{3}}, \end{array}$$
(12)$$a=1.57{\left\{{\left[0.6+0.321{\left(\frac{s}{D}\right)}^{-0.5}\right]}^{6}-0.083{\left(\frac{s}{D}\right)}^{-2.25}\right\}}^{\frac{1}{3}};\;0.12\leq \frac{s}{D}\leq 0.26,\;\mathrm{and}\;$$
(13)$${\tilde{g}}_{mod}=1-0.75{\left(\frac{D_{o}}{D}\right)}^{-0.3\left[0.12-4.2\;\ln\left(1-\frac{s}{D}\right)\right]}.$$

However, they found that Equation (12) was not an exact fit to their experimental results. Thus, they completed the correlation by introducing the correction factor *K*.
(14)$${\mathrm{Nu}}_{L}=K{\;\mathrm{Nu}}_{L,\;\mathrm{Eq}.\left(12\right)}$$
(15)$$K=-0.21{\left(\frac{D_{o}}{D}\right)}^{3}+1.40{\left(\frac{D_{o}}{D}\right)}^{2}-2.89\left(\frac{D_{o}}{D}\right)+2.72,\;\;1.2\leq \frac{D_{o}}{D}\leq 2.8$$

The correction factor might compensate for experimental errors that originate from various causes such as the accuracy of the instruments, response speed, and measurement errors. However, given the improvements in the accuracy of modern computational thermo-fluid dynamics, a numerical analysis can be used to avoid such errors as well as to reduce the labor, time, and costs of such experiments.

Accordingly, in this paper, we report on the application of the numerical method presented in Part 1 [1] by considering a single circular fin-tube heat exchanger model as a counterpart of the model in Kang and Chang [9]. Additionally, we analyzed the limiting cases in Figure 1 which were not elaborated upon by Kang and Chang [9], or the main idea presented in Equation (11) using the numerical technique. In this research, we proposed a corrected correlation based on the empirical result of Kang and Chang [9].

## 2. Heat Exchanger Model

The schematic of the circular fin-tube heat exchanger, the object of analysis in this study, appears in Figure 2. *D* and *D_o_* are the diameters of the circular fin and tube, and $P_{f}$, t, and s denote the pitch, thickness, and gap of the fins, respectively. This study expands the test cases from the original configurations of Kang and Chang [9] to 32 fin-tube combinations (see Table 2), and uses the commercial code ANSYS CFX 18 [10] for the numerical simulations. Unsteady and laminar flow conditions are assumed for the entire computation. The details of the numerical method are elucidated in Part 1 [1] of this paper.

## 3. Limiting Cases

### 3.1. Lowest Case: D_o_/D → 1, s/D = Finite Value (Single Horizontal Cylinder)

Figure 3 explains how the computational cases regarded in this work assume the lowest limiting case (see Figure 1), where the diameter ratios (*D_o_*/*D*) of D12, D11, D10, … (the numbers following D mean ten times the value of *D_o_*/*D*; thus, 1.20, 1.07, 1.01, …) continue to decrease gradually to approach the limiting case of *D_o_*/*D* = 1.0. Then, 12 types of heat exchangers were numerically analyzed for the fin pitches P12, P17, P21, and P26.

In Figure 4, Nu, based on the characteristic length $L=\pi \left(D+D_{o}\right)/4,$ is plotted versus Ra, which is based on the gap size of the fins (${\mathrm{Ra}}_{s}$). The graphs for the normalized fin pitches P12 (minimum fin pitch) and P26 (maximum fin pitch) are plotted with dashed and solid lines, respectively, as representative examples. As Ra increases, so does Nu. This trend was expected because of the enhanced natural convection. The two groups of graphs in Figure 4 can be distinguished with regard to discontinuity, but the slopes of D10, D11, and D12 are assessed as the *n*th power of Ra, namely, the powers of 0.22, 0.23, and 0.26, respectively. Consequently, the power seems not to converge exactly to 0.25, unlike the case of Figure 1; however, it approaches a finite value of less than 0.22 (which is 12% lower than 0.25).

In Figure 5, the characteristic lengths of Nu and Ra are changed to the tube diameter D, and the results correspond to the upper limit of the heat transfer in the case of a single horizontal cylinder. In this range of experiments, the existence of the fin creates an adverse effect, namely, a heat transfer deficit for the same pitch with taller fins or a higher value of *D*. The opposite effect is obvious for shorter fins or a lower *D*. As the fin pitch increases, or as P rises, the heat transfer reaches the limiting case.

### 3.2. Hightest Case: D_o_/D → ∞, s/D << 1 (Vertical Parallel Disks)

Figure 6 presents the fin models corresponding to the highest limiting case in Figure 1. To approximate the geometry with vertical flat plates, the fin diameter is increased (with D28, D50, D100, … corresponding to $\frac{D_{o}}{D}=$2.8, 5.0, 10.0, …), and 12 types of heat exchangers were numerically analyzed for a wide range of fin pitches (namely, P12, P17, P21, and P26).

In Figure 7, ${\mathrm{Nu}}_{L}$ is plotted against ${\mathrm{Ra}}_{s}$ (see Figure 4) for the lowest case. Moreover, the results of the minimum and maximum fin gaps, P12 and P26, are denoted via dashed and solid lines, respectively. Unlike Figure 4, the curves seem to approximate to a line even with the variation in the pitch, and the slopes increase for ascending diameters (0.51, 0.63, and 0.78). It was not possible to reach the value of unity (n=1) due to the load limitation for the computational domain. Nonetheless, the trend in Figure 8 shows that it can converge to 1.0 at $D/D_{o}\to \infty$. Figure 8 also shows the possibility of the lowest limit (n=0.25 when $D/D_{o}=1$) despite some amount of error.

## 4. Classification Criteria for Types A and B

The results of the analyses showing Nu versus Ra and the categorization of the two types are presented in Figure 9. Type A shows separated curves while type B is described by linkages of the enveloped curves for each diameter ratio. For example, D10–D12 can be categorized as type A, and D18, D22, and D26 can be categorized as type B. To present the bounds of the two types for D15 or $D_{o}/D=1.5$ as an example, Table 3 is presented as an expansion of the parametric list in Table 2. The same numerical method is used in the analysis.

Therefore, the boundary at $D_{o}/D=1.2$ serves to classify the two types, as seen below:(16)$$\mathrm{Type}\;A\leq D_{o}/D=1.2<\mathrm{Type}\;B$$

For $D_{o}/D\leq 1.2$ (short fins), the curves denote type A, whereas for $D_{o}/D>1.2$ (tall fins), the curves denote type B. The parametric range can be specified as $1.0<D_{o}/D\leq 10$ and 0.12≤s/D<0.26. As commented in Section 3.1, $D_{o}/D\to 1$ converges to a circular cylindrical model.

## 5. Correlation Expansion and Validity

### 5.1. Expansion of Correlation

The classical correlations of natural convection on a single cylindrical tube follow the relationship $C{{\mathrm{Ra}}_{D}}^{n}$, similarly to Equations (1)–(4), which are most commonly used in natural convection research. Some consideration should also be given to the application of these types of correlations to the proposed models (circular fin-tube heat exchangers). Type A in the previous section appears to best fit the circular tube correlation of Morgan [3]. However, type B is affected by the fin diameter, and the gaps between the fins must also be considered in the correlation to ensure that the correct characteristic lengths are selected, in line with the conventions of physics. The errors of the results of the computational fluid dynamics presented in this paper against the correlation of Kang and Chang [9] are shown in Table 4. For D15–D18, the mean errors are limited to 18.5%, namely within the error levels reported in Part 1 [1]. However, for the limiting cases of D50 and D100, the errors tend to deviate to a greater extent because these diameter ratios do not fit within the correlation range of Kang and Chang [9]. Therefore, in this research, we also present an expanded version of the correlation for type B.

Using the data of D15–D100, regressed lines are extracted in Figure 10 to the form $C{\mathrm{Ra}}_{s}{}^{n}$ so as to express the correlations as in the case of a circular tube. However, recall that the characteristic length for Ra is equal to the gap of the fins s instead of the tube diameter D.

In Figure 11, the proportional coefficient C is a function of the diameter ratio ($D_{o}/D$), which decreases in proportion to the −1.175 power of $D_{o}/D$ within an $R^{2}$ variance of 97% for the computational data.
(17)$$C=1.76{\left(\frac{D_{o}}{D}\right)}^{-1.175}.$$

In Figure 12, the power n is the logarithmic function of $D_{o}/D$ within an $R^{2}$ variance of 97% for the computational data.
(18)$$n=0.2+0.262\ln\left(\frac{D_{o}}{D}\right).$$

Thus, the final correlation is summarized from Equations (17) and (18) as follows:(19)$${\mathrm{Nu}}_{L}=C{\mathrm{Ra}}_{s}{}^{n},5<{\mathrm{Ra}}_{s}<200\;\mathrm{and}$$
(20)$$1.2<\frac{D_{o}}{D}\leq 10,\;0.12\leq \frac{s}{D}<0.26.$$
where C and n are defined in Equations (17) and (18), respectively, and Equation (20) provides the ranges of the parameters, which are considerable extensions of the work completed by Kang and Chang [9].

### 5.2. Validity of Correlation

The validity of the correlation, expressed in Equation (19), is checked in Figure 13, where Nu is compared the numerical data for the simultaneous Ra. It was found that 86.4% of the entire data are included within 10% of the region bounded by the red solid lines, and all of the data are located inside 15% of the region bounded by the blue dashed lines.

Figure 14 shows a comparison of the present correlation with the experimental data in Kang and Chang [9]. Overall, the correlation was predicted downward from the experimental data but trends were similar. For (a), (b) and (c), there are errors of up to 24.9%, 17.7%, and 20.1%, respectively. These results are due to the fact that the computational data underpredict experimental data in Kang and Chang [9] by approximately 16% - 20%. This reason has already been mentioned in Part 1 [1].

## 6. Conclusions

In this research, 36 types of circular fin-tube heat exchanger models were numerically studied to analyze the effect of shape parameters such as tube diameter, fin diameter, and gap of fins. The following conclusions were arrived at by analyzing the data to an extended version of Kang and Chang’s correlation [9]:

We considered two limiting conditions. The lower limit (*D_o_* /*D* → 1, *s*/*D* = finite value) corresponds to a horizontal circular tube, while the upper limit (*D_o_*/*D* → ∞, *s*/*D* << 1) corresponds to vertical flat plates. The main idea of the empirical correlation proposed by Kang and Chang [9] was verified using extended parameters, as the experiment could not cover these conditions. The power of Ra (which is based on the gap of the fins) proportional to Nu was computed as 0.22 at a minimum ($D_{o}/D=1.01$) and 0.78 at a maximum ($D_{o}/D=10.0$). Although these values differ from the theoretical results of Kang and Chang’s correlation [9], they show the possibility of using the numerical analysis for prediction over a far wider range of parameters.

The parametric plane was divided into two types: type A, where all the curve groups of variable pitches are clearly separated from one another, and type B, where all the curves meet on each envelop for each diameter ratio group. The separating boundary for the criteria is depicted by $D_{o}/D=1.2$ (i.e., a diameter ratio less than this value (or short fins) will be classified as type A, whereas that greater than this value (or tall fins) will be categorized as type B).

Using the computational fluid dynamics technique for the investigation of the limiting conditions allowed us to considerably extend the parametric range for type B in this research to 1.2 < $D_{o}/D$ ≤ 10, and the complex correlation was simplified in the form ${\mathrm{Nu}}_{L}$= $C{\mathrm{Ra}}_{s}{}^{n}$, where C and n are the functions of the diameter ratio ($D_{o}/D$). Moreover, approximately 87% of the computational data lie within the 10% error range when compared with the empirical correlation.

The ranges of parameters in this research are defined as ${5<\mathrm{Ra}}_{s}<200$, 1.2 < $D_{o}/D$ ≤ 10, and 0.12≤s/D<0.26, and such wide bandwidths can be applied to various circular fin-tube heat exchangers in practice.

## Figures and Tables

**Figure 1 entropy-22-00358-f001:**
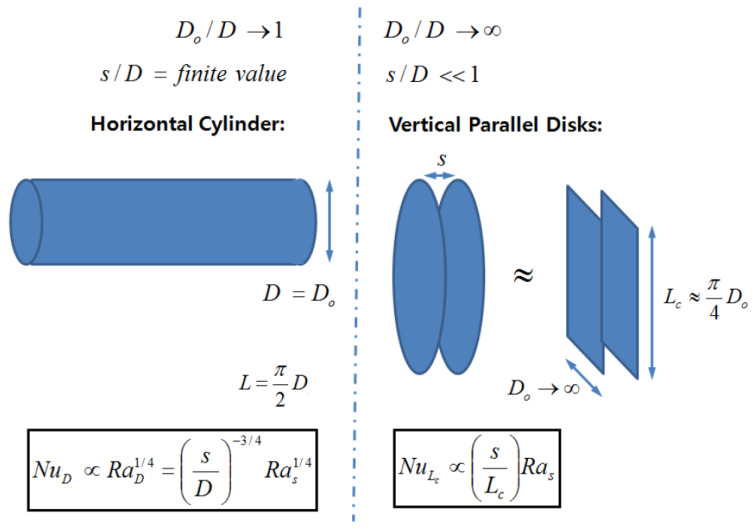
Primary regime between two extreme conditions for natural convection in the circular fin-tube configuration. (from Kang & Chang [9]).

**Figure 2 entropy-22-00358-f002:**
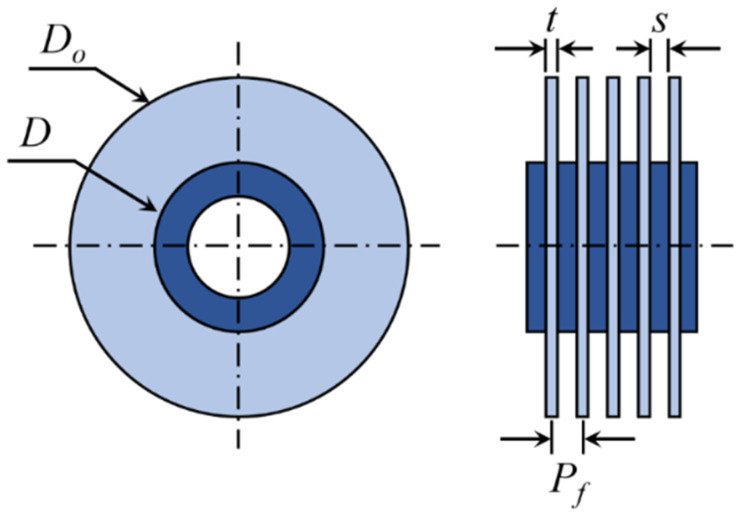
Schematic diagram of circular fin-tube heat exchanger studied in the present work.

**Figure 3 entropy-22-00358-f003:**
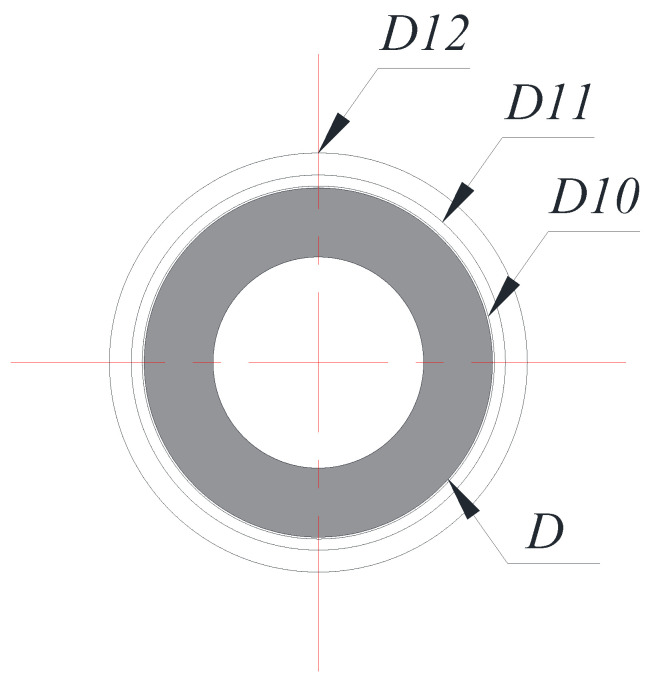
Schematic diagram of lowest cases on limit model.

**Figure 4 entropy-22-00358-f004:**
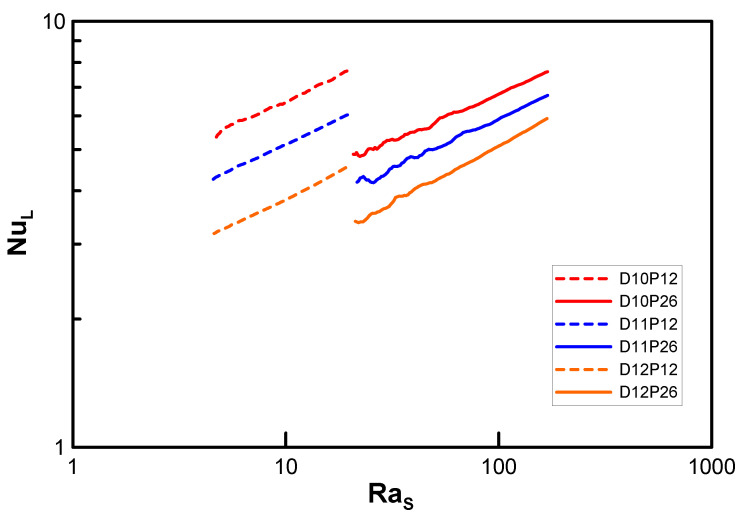
Variation of Nusselt number with Ra_s_ in the lowest limit model.

**Figure 5 entropy-22-00358-f005:**
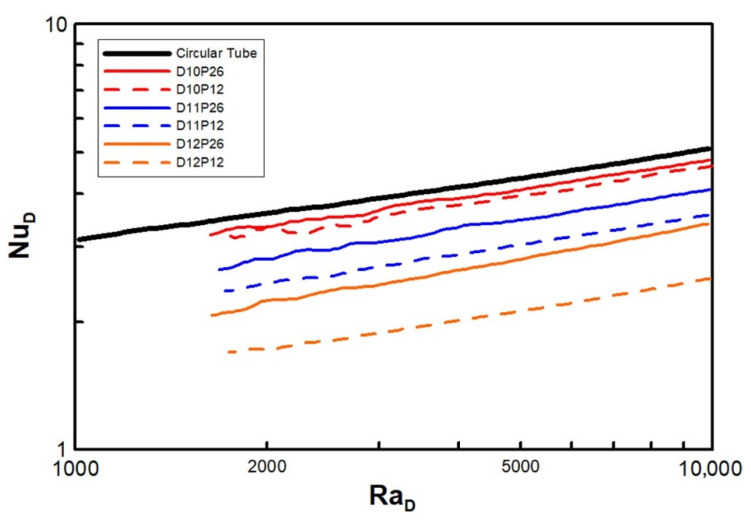
Variation of Nusselt number with Ra*_D_*.

**Figure 6 entropy-22-00358-f006:**
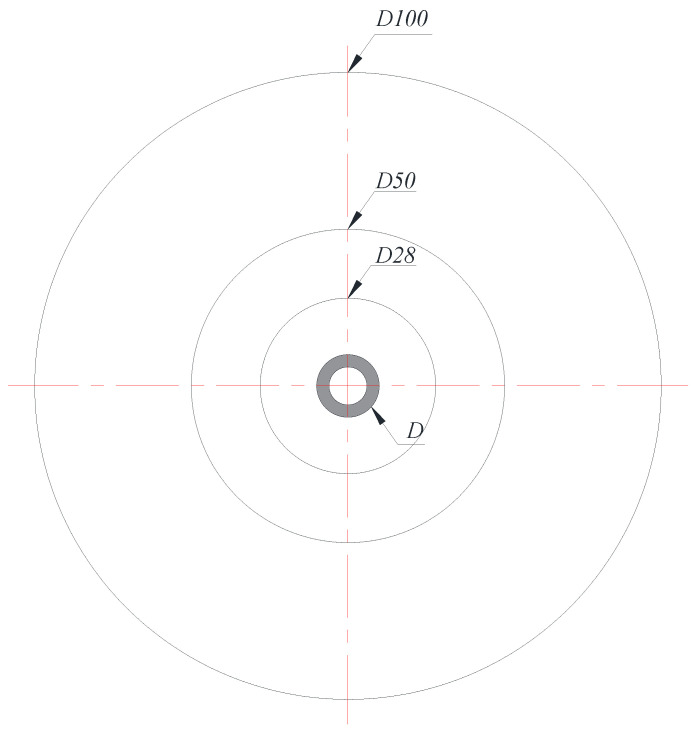
Schematic diagram of highest cases on limit model.

**Figure 7 entropy-22-00358-f007:**
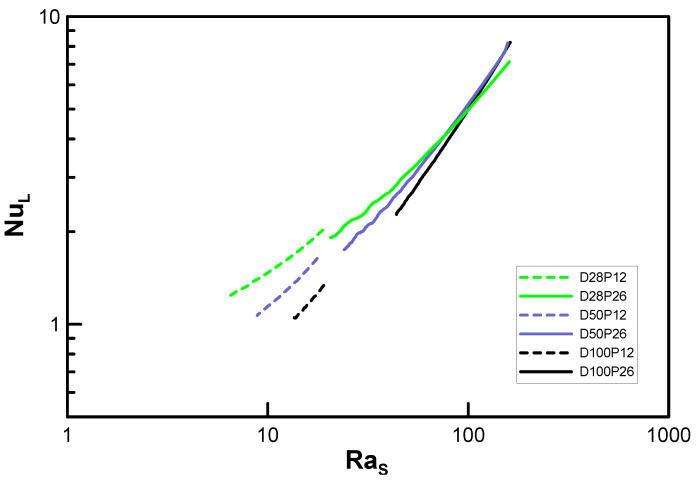
Variation of Nusselt number with Ra_s_ in the highest limit model.

**Figure 8 entropy-22-00358-f008:**
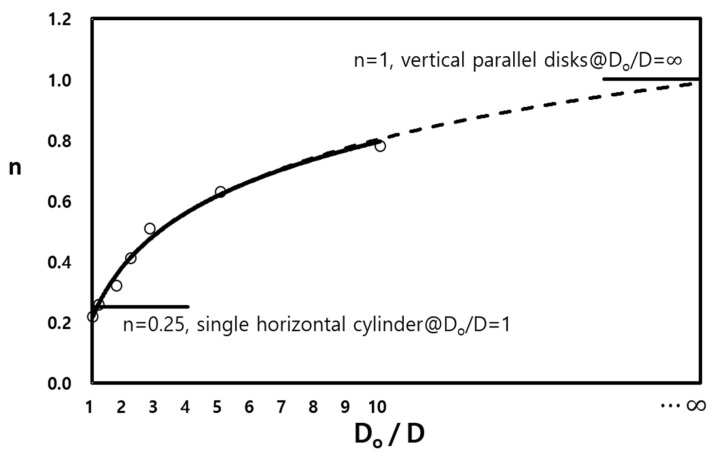
The value of the power *n* according to the diameter ratio of the circular fin-tube.

**Figure 9 entropy-22-00358-f009:**
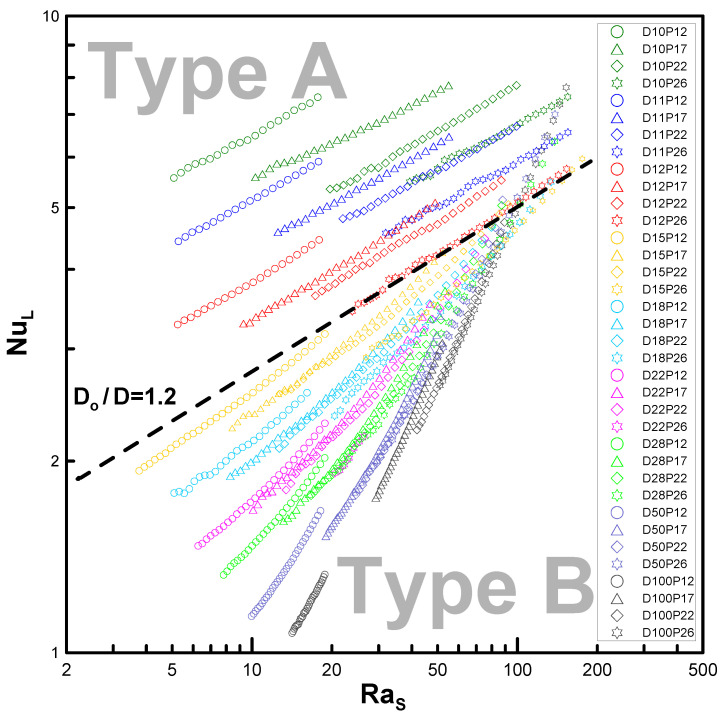
Variation of Nusselt number with ${\mathrm{Ra}}_{s}$: Classification criteria for type A and B.

**Figure 10 entropy-22-00358-f010:**
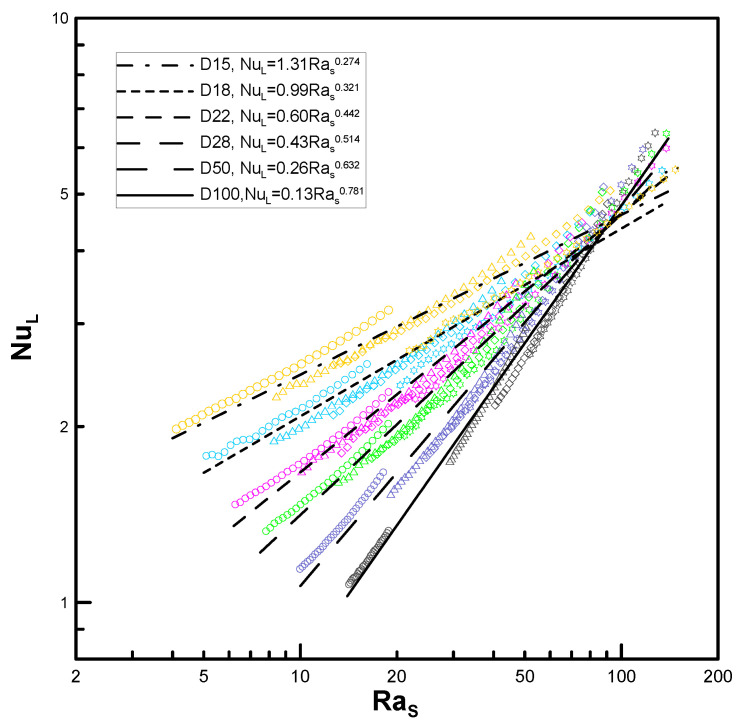
Variation of Nusselt number and trend line with ${\mathrm{Ra}}_{s}$.

**Figure 11 entropy-22-00358-f011:**
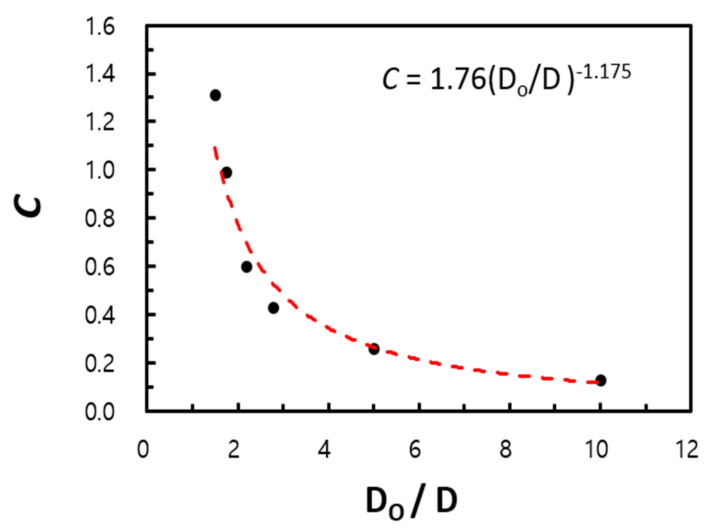
Variation of C value with circular fin-tube diameter ratio.

**Figure 12 entropy-22-00358-f012:**
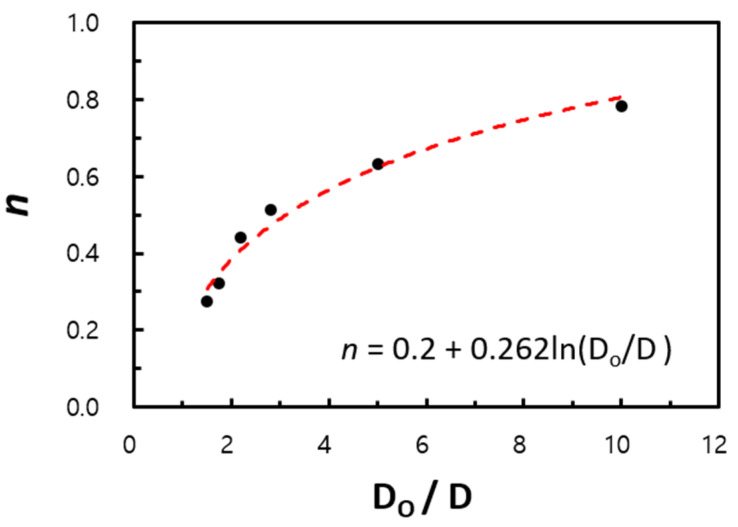
Variation of n value with circular fin-tube diameter ratio.

**Figure 13 entropy-22-00358-f013:**
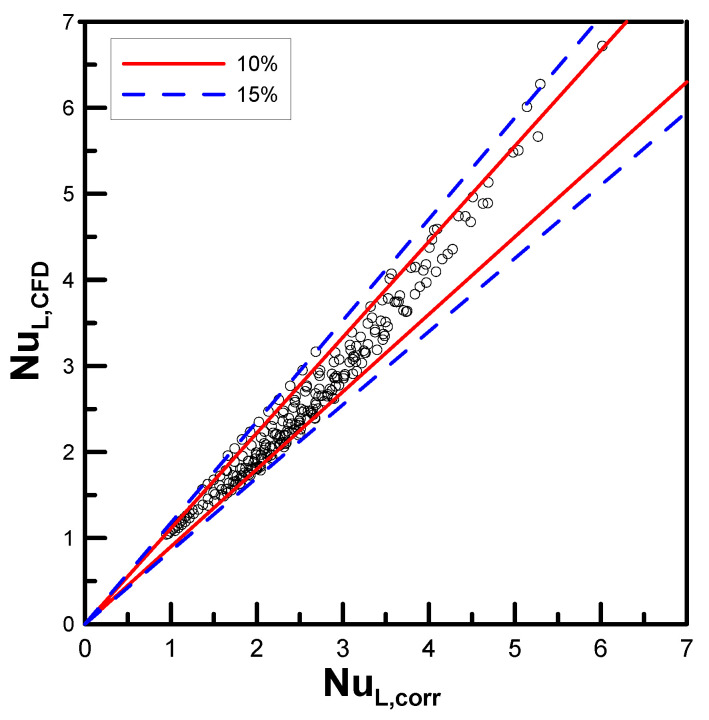
Comparison of the present correlation with the numerical data.

**Figure 14 entropy-22-00358-f014:**
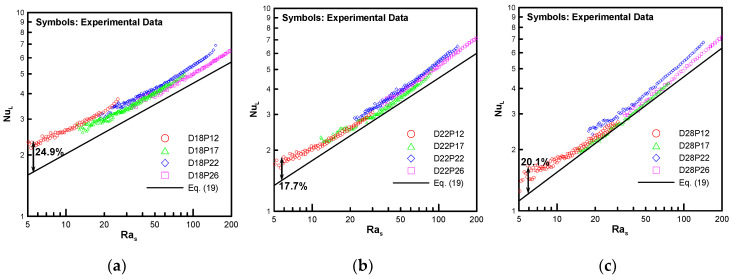
Comparison of the present correlation with experimental data: (**a**) D18, (**b**) D22, and (**c**) D28.

**Table 1 entropy-22-00358-t001:** The range of Rayleigh numbers, C and n values used in the correlation of Morgan [3].

${\mathrm{Ra}}_{D}$	C	n
$10^{-10}\sim 10^{-2}$	0.675	0.058
$10^{-2}\sim 10^{2}$	1.020	0.148
$10^{2}\sim 10^{4}$	0.850	0.188
$10^{4}\sim 10^{7}$	0.480	0.250
$10^{7}\sim 10^{12}$	0.125	0.333

**Table 2 entropy-22-00358-t002:** Dimensions of the fin-tube heat exchangers tested in the present work.

Case		*D*	*D_o_*	*P_f_*	*t*	*D_o_*/*D*	*s*/*D*	Case		*D*	*D_o_*	*P_f_*	*t*	*D_o_*/*D*	*s*/*D*
**D10**	**P12**	15.88	16.1	2.89	1.0	1.01	0.119	**D22**	**P12**	15.88	34.9	2.89	1.0	2.20	0.119
	**P17**			3.68			0.169		**P17**			3.68			0.169
	**P21**			4.26			0.205		**P21**			4.26			0.205
	**P26**			5.06			0.256		**P26**			5.06			0.256
**D11**	**P12**		17.1	2.89		1.07	0.119	**D28**	**P12**		44.5	2.89		2.80	0.119
	**P17**			3.68			0.169		**P17**			3.68			0.169
	**P21**			4.26			0.205		**P21**			4.26			0.205
	**P26**			5.06			0.256		**P26**			5.06			0.256
**D12**	**P12**		19.1	2.89		1.20	0.119	**D50**	**P12**		79.4	2.89		5.00	0.119
	**P17**			3.68			0.169		**P17**			3.68			0.169
	**P21**			4.26			0.205		**P21**			4.26			0.205
	**P26**			5.06			0.256		**P26**			5.06			0.256
**D18**	**P12**		27.8	2.89		1.75	0.119	**D100**	**P12**		158.8	2.89		10.0	0.119
	**P17**			3.68			0.169		**P17**			3.68			0.169
	**P21**			4.26			0.205		**P21**			4.26			0.205
	**P26**			5.06			0.256		**P26**			5.06			0.256

**Table 3 entropy-22-00358-t003:** Dimensions of D15 heat exchanger added for classification.

Case		*D*	*D_o_*	*P_f_*	*t*	*D_o_*/*D*	*s*/*D*
D15	P12	15.88	23.8	2.89	1.0	1.50	0.119
	P17			3.68			0.169
	P21			4.26			0.205
	P26			5.06			0.256

**Table 4 entropy-22-00358-t004:** Error value between B type heat exchanger and Kang and Chang’s correlation [9].

Case		$\left(1-\frac{{\mathrm{Nu}}_{CFD}}{{\mathrm{Nu}}_{K\&C\;corr}}\right)\times 100,\;\%$			Case		$\left(1-\frac{{\mathrm{Nu}}_{CFD}}{{\mathrm{Nu}}_{K\&C\;corr}}\right)\times 100,\%$		
		Min.	Max.	Ave.			Min.	Max.	Ave.
D15	P12	15.2	20.8	18.3	D28	P12	0.0	22.3	12.7
	P17	6.3	13.9	11.6		P17	0.2	17.0	6.5
	P21	3.4	9.7	8.1		P21	0.2	17.2	7.1
	P26	7.7	14.2	12.4		P26	0.3	21.2	14.4
D18	P12	0.0	6.6	1.1	D50	P12	136.7	154.1	143.0
	P17	2.9	16.0	12.0		P17	143.7	165.0	150.8
	P21	0.0	13.0	9.0		P21	142.8	165.4	149.5
	P26	4.6	15.1	11.6		P26	138.3	156.6	143.2
D22	P12	6.3	21.7	16.5	D100	P12	101.7	102.2	101.9
	P17	0.0	17.3	11.4		P17	102.6	103.7	103.1
	P21	0.0	17.6	11.1		P21	102.6	103.7	103.0
	P26	0.0	19.4	13.9		P26	102.1	103.2	102.5

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
