# Peer review of "Numerical Analysis on Natural Convection Heat Transfer in a Single Circular Fin-Tube Heat Exchanger (Part 2): Correlations for Limiting Cases"

_entropy, 2020, doi:10.3390/e22030358_

Round 1

Reviewer 1 Report

This work proposed a corrected correlation based on the numerical results obtained for the natural convection heat transfer in a circular fin-tube heat exchanger. The proposed correlation also extended validity range comparing with the experimental correlation obtained by Kang and Chang [1]. The proposed correlation was validated with the experimental correlation obtained by Kang and Chang.

This contribution of this work is that it provides a validated, wider range, and simple correlation that can be used to rapidly evaluate the natural circulation heat transfer performance of this type heat exchanger. Such correlation can also be useful in the system-level modeling and simulation.

To the reviewer, the overall quality of this work is good. I would recommend the publication of this work in the journal of Entropy.

Several comments:

  1. The authors should include a reference to Part 1.
  2. Line 172-173, it is said “Type A in the previous section appears to best fit the circular tube correlation.” Please specify the correlation mentioned.
  3. For the validation, the reviewer would strongly suggest to further compare the proposed correlation (I.e., Eq. 19) with experimental data directly, by plotting the proposed correlation and experimental data in the same figure, for D12, D18, D22 and D28, respectively. Refer to Figure 7 in ref. [1] for the comparison. This would add value to the work by demonstrating directly how good the proposed correlation will be.

Reviewer 2 Report

I would like to thank the editors for giving me the opportunity to review this interesting work.

The authors quantified the relationship between the geometry and heat transfer rating of a finned-tube heat exchanger, by means of determining two separate rating zones depending on geometrical parameters such as fin spacing, diameter and tube diameter. The authors revealed the two rating zones where different natural convection heat transfer mechanisms occur. However, the trends of graphs have not been discussed thoroughly from the scientific conclusion perspective.

General Revisions: Nomenclature is missing. Table 2 is missing.

Line 13: What the two limiting conditions(or the types A and B) are is not clear in the abstract

Line 27: "This far", instead of "Thus far"

Line 33: Instead of "introducing C and n", "tabulating C and n" sounds more reasonable

Line 37: The relationship of the C and n with the Prandtl number is not visible from the Table 1, despite the fact that it was stated to be related at the Line 34.

Line 87: Use Kang and Chang [1] instead of the expression "Ref. [1]"

Line 180 and 181: Give references of Kang and Chang

Line 218 and 222: Missing Kang and Chang reference
